# Monitoring and modeling infiltration-recharge dynamics of managed aquifer recharge with desalinated seawater

Yonatan Ganot[1,2], Ran Holtzman[2], Noam Weisbrod[3], Ido Nitzan[1], Yoram Katz[4] and Daniel Kurtzman[1]

[1]Institute of Soil, Water and Environmental Sciences, The Volcani Center, Agricultural Research Organization, Rishon LeZion, 7528809, Israel
[2]Department of Soil and Water Sciences, The Hebrew University of Jerusalem, Rehovot, 7610001, Israel
[3]Department of Environmental Hydrology & Microbiology, Zuckerberg Institute for Water Research, Jacob Blaustein Institutes for Desert Research, Ben-Gurion University of the Negev, Midreshet Ben-Gurion, 8499000, Israel
[4]Mekorot, Water Company Ltd, Tel Aviv, 6713402, Israel

*Correspondence to*: Yonatan Ganot (yonatan.ganot@mail.huji.ac.il)

**Abstract.** We study the relation between surface infiltration and groundwater recharge during managed aquifer recharge (MAR) with desalinated seawater in an infiltration pond, at the Menashe site that overlies the northern part of the Israeli Coastal Aquifer. We monitor infiltration dynamics at multiple scales (up to the scale of the entire pond) by measuring the ponding depth, sediment water content and groundwater levels, using pressure sensors, single-ring infiltrometers, soil sensors and observation wells. During a month (January 2015) of continuous intensive MAR ($2.45 \cdot 10^6$ m$^3$ discharged to a 10.7 hectare area), groundwater level has risen by 17 m attaining full connection with the pond, while average infiltration rates declined by almost 2 orders of magnitude (from ~11 to ~0.4 m d$^{-1}$). This reduction can be explained solely by the lithology of the unsaturated zone that includes relatively low-permeability sediments. Clogging processes at the pond-surface—abundant in many MAR operations—are negated by the high-quality desalinated seawater (turbidity ~0.2 NTU, total dissolved solids ~120 mg L$^{-1}$) or negligible compared to the low-permeability layers. Recharge during infiltration was estimated reasonably well by simple analytical models, whereas a numerical model was used for estimating groundwater recharge after the end of infiltration. It was found that a calibrated numerical model with a one-dimensional representative sediment profile is able to capture MAR dynamics, including temporal reduction of infiltration rates, drainage and groundwater recharge. Measured infiltration rates of an independent MAR event (January 2016) fitted well to those calculated by the calibrated numerical model, showing the model validity. The successful quantification methodologies of the temporal groundwater recharge are useful for MAR practitioners and can serve as an input for groundwater flow models.

## 1 Introduction

Managed aquifer recharge (MAR) is a common practice in water resources management in which excess water is stored in the aquifers for future consumption. Major techniques used for aquifer recharge include well injection, bank filtration, rainwater harvesting and infiltration ponds (Dillon, 2005). In the Israeli Coastal Aquifer, MAR started in 1958 with Lake Kinneret water, surface runoff and carbonate-aquifer groundwater as the recharge sources (Sellinger and Aberbach, 1973).

Between the years 2000 to 2013 MAR in the Israeli Coastal Aquifer was mostly (88%) from the soil aquifer treatment ponds at the Shafdan sites where secondary effluents are delivered into infiltration ponds for tertiary treatment. The remaining MAR can be primarily attributed to storm runoff according to the seasonal rainfall (Israel Hydrological Service, 2013). Recently, desalinated seawater—a relatively new water source in Israel (Stanhill et al., 2015)—has been occasionally used as source water for MAR.

MAR using desalinated seawater (DSW) poses several scientific and operative challenges due to the unique water composition compared to natural groundwater. Yet, scientific publications on MAR with DSW are few. Field tests of MAR using DSW were performed during the 1970s and the 1990s in clastic and carbonate aquifers in Kuwait. Well clogging was identified as a major concern, especially in the clastic aquifers (Mukhopadhyay et al., 1994). These field tests were followed by laboratory studies focusing on clogging and geochemical processes using core experiments with DSW (Al-Awadi et al., 1995; Mukhopadhyay et al., 1998, 2004). A closely related study, on MAR with reverse-osmosis wastewater was conducted at the St-André MAR site in Belgium. Reported work includes flow and transport modeling (Vandenbohede et al., 2008, 2009a; Vandenbohede and Van Houtte, 2012), isotope and geochemical analysis (Kloppmann et al., 2008; Vandenbohede et al., 2009b) and reactive transport modeling (Vandenbohede et al., 2013).

In this paper we focus on infiltration and recharge dynamics during MAR with DSW. This work is part of a comprehensive study involving field and laboratory investigations in order to better understand hydrological and geochemical processes during MAR with surplus of reverse-osmosis DSW in Israel (Ronen-Eliraz et al., 2017). The geochemical perspective of this field study will be reported in a future publication (Ganot et al., 2017). The results reported here are unique for several reasons. First, we monitored a month (January 2015) of continuous MAR with $2.45 \cdot 10^6$ m$^3$ of DSW (loading of about 23 m month$^{-1}$), higher than in most other reported MAR at infiltration basins, comparable only to few studies (Kennedy et al., 2014; Nadav et al., 2012; Racz et al., 2012). Second, we focus on the temporal pond-surface infiltration and groundwater recharge using field measurements and both simplified analytical methods as well as detailed numerical modeling. Third, for the numerical model we use measured data for variable-head boundary conditions at both top and bottom boundaries. Our data allows us to specifically address the lag between infiltration and groundwater recharge. This is of interest in cases of relatively deep unsaturated zone (initially ~25 m, in this study) in order to better estimate groundwater recharge. In contrast, conventional methods used in many studies estimate potential groundwater recharge from infiltration rates that do not necessarily represent recharge rates at the water table (Scanlon et al., 2002).

The purpose of this paper is to provide a detailed field-scale analysis of MAR with DSW from a hydrological perspective. Initially the monitoring system is described, the unsaturated zone is characterized and several methods for calculating infiltration and recharge are presented. Next, infiltration dynamics (spatial and temporal) and its relation to the unsaturated zone lithology and to pond surface clogging is discussed. Finally, groundwater recharge estimations obtained from the analytical and numerical models are evaluated and compared.

## 2 Methods

### 2.1 Site description

The Menashe MAR site is located on sand dunes, 28 m above mean sea level (AMSL), overlaying the northern part of the Israeli Coastal Aquifer (Fig. 1a). Climate is Mediterranean with annual mean precipitation of 566 mm yr$^{-1}$ (Gan Shmuel, 1987–2007). The annual average temperature is 20.2 °C. The coldest month is January with average maximum and minimum temperatures of 17.4 and 10.5 °C, respectively. The warmest month is August with average maximum and minimum temperatures of 28.5 and 24.8 °C, respectively (Israel Meteorological Service, 2016). Below the Menashe MAR site the aquifer is about 80 m deep and consists mainly of calcareous sandstone (Kurkar hereafter), sand, silty mud and clay lenses. Regional groundwater level is ~3 m AMSL (September 2014), and pre-winter to post-winter seasonal groundwater-level fluctuations are ~2 m (Israel Hydrological Service, 2014). Characteristic hydraulic properties of the aquifer are 10 m d$^{-1}$, 0.25 and 0.4, for hydraulic conductivity, storativity and porosity, respectively (Shavit and Furman, 2001). Operating since 1967, the Menashe MAR site diverts the natural ephemeral flow into a settling pond and from there to three infiltration ponds. The recharged water is recovered from the aquifer by dedicated production wells that encircle the site (Sellinger and Aberbach, 1973). In the vast majority of runoff events only the two northern infiltration ponds are used. Therefore, in the last few years, the southern infiltration pond is used for infiltration of surplus of DSW from the Hadera reverse-osmosis desalination plant, located 4 km to the west on the coast (Fig. 1b).

### 2.2 Monitoring

Monitoring of MAR activity was performed in the southern infiltration pond (herein referred to as 'the pond') where DSW discharged occasionally according to operational considerations of Mekorot (Israel national water company) and the Israeli Water Authority. A dedicated monitoring system including observation wells, soil sensors and infiltration rings was installed at the south part of the pond (Fig. 1c, e). The two groundwater observation wells (OA and OB) are 30 m deep, perforated at the lower part of the well (10 m from the bottom) and penetrating the saturated zone. Both were monitored by loggers (CTD-Diver, Eijkelkamp) measuring pressure head and electrical conductivity (EC). The shallow unsaturated zone includes 8 soil sensors (5TE and GS3, Decagon Devices) at depths of 0.3, 0.5, 1, 1.5, 2, 2.5, 3 and 4 m below the pond surface, measuring volumetric water content (WC) and bulk EC (a measure of the electric conductivity of the bulk soil, which includes soil, water and air). The monitoring system continuously operated since October 2014 and measurements were obtained regularly every 15–30 min and at a finer resolution of 1–5 min during MAR or infiltration tests. In addition to the permanent monitoring system, ponding depth was monitored by three pressure loggers installed on the pond surface for the January 2015 MAR event at the north, center and south part of the pond (Fig. 1c). All pressure head measurements were compensated by on-site logging barometer (BARO-Diver, Eijkelkamp).

### 2.3 Sediment sampling

Disturbed sediment samples (from auger) were taken during the drilling of the observation wells. Relatively undisturbed continuous core samples from the unsaturated zone were obtained by a direct-push rig (9700-VTR PowerProbe, AMS). Cores were taken at the following locations: next to the soil sensors (0–12 m depth), the southern road (0–9 m depth) and east of the pond (0–6 m depth, Fig. 1d). All sediment samples (both disturbed and undisturbed) were analyzed for particle size distribution by sieving for gravel (>2 mm), sand (2–0.045 mm) and hydrometer for silt and clay. Bulk densities were calculated only for the undisturbed samples from the mass and volume of the cores and their water content.

The sediment profiles of observation wells OA and OB are similar. The top 30 m includes two repeated sequences, each consisting of a sand layer overlaying a sandy-clay-loam (SCL) layer, down to ~15 m, of variable depth and thickness. Deeper down the profile, Kurkar is dominant, alternating with layers of sand and sandy loam. The shallower direct-push profile next to the soil sensors location (Ds) is also similar to the profiles of OA and OB (due to their proximity to each other – less than 15 m apart), while the more distant profiles (Dr and De) are less similar (Fig. 1d). We cannot determine the lateral extent of the less-permeable layers based on our sediment sampling, which is spatially limited. We assume that these clayey and loamy layers are discontinuous like most of the low-permeability lenses at distances greater than 3 km from the coastline in the Israeli Coastal Aquifer (Kurtzman et al., 2012).

### 2.4 Calculating infiltration rates

Infiltration rates were calculated at the pond-scale by pond draining rate, and at local-scale by single-ring infiltrometers and wetting-front propagation (Dahan et al., 2007). Details of each method are given below in the following sub-sections.

#### 2.4.1 Pond scale

Ponding depth data was used to calculate the pond-scale infiltration rates. This method represents the infiltration rate of the whole pond, which is an average of the local infiltration rates (that may vary spatially due to sediment heterogeneity). The average pond infiltration rates were calculated by linear-regression of ponding-depth, which declined due to intermittent inlet discharge during the January 2015 MAR event. Each observation point of infiltration-rate was calculated from a large number (tens to hundreds) of ponding-depth measurements. Two conditions must be met in order to calculate infiltration rates by this method: (1) ponding depth is declining solely due to infiltration (i.e., no other inlet/outlet source or surface flow) and (2) the time span of the descending ponding-depth data is sufficiently long (usually at least few hours) in order to obtain regression with low-error slope (which is an estimate of the pond-scale infiltration rate).

#### 2.4.2 Single-ring infiltrometers

In order to capture local infiltration-rate variability, we used an array of 24 single-ring infiltrometers (100 cm long, 20 cm diameter) hammered 60 cm into the ground at different locations (Fig. 1e). Sediment samples taken from each infiltration

ring location (outside the ring) at depth of 5 cm (undisturbed) and 60 cm (disturbed, with an auger, divided to 4 sections) were analyzed for bulk density (undisturbed only) and particle size distribution (both).

Infiltration tests were performed under relatively dry conditions (average WC of 0.09 $m^3 m^{-3}$), early ponding (4 hours after MAR started) and late ponding (after 1 month when discharge into the pond was ended), by continuously monitoring water level inside the rings. Under dry conditions, a fixed volume of 5.1 L was used in each ring to ensure one dimensional (1D) flow inside the single-ring. The infiltration rates were calculated by linear-regression of pressure-head vs. time data, for dry conditions (n=24), early ponding (n=11) and late ponding (n=20). Infiltration tests with relative errors higher than 15% were omitted from the analysis. The omitted results (6 out of 61 single-ring tests) were due to insufficient measurement-points during the infiltration test. We found the single-ring method more suitable than other methods we tested (double ring and Guelph permeameter); mainly because of its simplicity and permanent location that allows infiltration test repetitions under different conditions.

### 2.4.2 wetting-front propagation

Monitoring of water-content variation in the unsaturated zone by the soil sensors provides information on the wetting/drying-front propagation velocity. Infiltration (or drainage) rates are evident from the lag in wetting (or drying) front between different sensors at various depths. Infiltration rates were estimated from the velocity of the fronts and the difference in water-content on both sides of the wetting front.

### 2.5 Modeling groundwater recharge

We describe the flow from the surface to the water table during a MAR operation using three different models: two simple analytical models (i.e., one using water-table data, the other ponding-depth data) and one numerical model (in which both data sets were used). The simple analytical models are useful not only when there is not enough data to calibrate a numerical model; they also provide a first approximation which can be used as a preliminary test for a numerical model. In all three models we assign similar sediment profile layers and saturated hydraulic conductivity (Table 1), evaluated from pedotransfer functions (PTF) using bulk density and particle size distribution data (Schaap et al., 2001). Only the saturated hydraulic conductivity of the top SCL layer was modified during calibration of the numerical model. Temporal and cumulative infiltration/recharge were obtained using 5 min resolution data measured during the January 2015 MAR event.

Since the pond water depth is much smaller than its horizontal dimensions, we consider 1D vertical infiltration (perpendicular to the layers), e.g. see Philip (1992). This assumption neglects lateral water flow, which is mainly relevant at the pond boundaries and during early and late stages of MAR (when only a portion of the pond surface is covered with water). However, during most of the January 2015 MAR event the whole pond area was covered with water and therefore the 1D flow is a reasonable approximation. A main advantage of 1D model, apart from its simplicity, is that it can capture the whole-pond MAR processes by a single representative 1D sediment profile.

### 2.5.1 Analytical lumped model using hydraulic conductivity and water table data

In this lumped model, we use two measured data sets, water table levels and saturated hydraulic conductivities. We consider the following transient boundary conditions: flux at the top and water table level (head) at the bottom. At the top, the recharge flux is equivalent to the flux in a saturated layered column under unit gradient flow. The lower boundary is the level of the moving water table. Calculations begin from the time that the water table starts rising, with initial conditions of fully saturated sediment profile. Assuming initially saturated profile allows using the saturated hydraulic conductivity ($K_s$). We consider the flux $q$ [L T$^{-1}$] to be equal to the equivalent vertical hydraulic conductivity $\overline{K_s}$ of the layers above the water-table, which, for a vertical flow perpendicular to the layers, is computed from the harmonic mean of the layers saturated hydraulic conductivities

$$q = \overline{K_s} = L / \sum (z_i / K_{s,i}) \tag{1}$$

where, $z_i$ [L] and $K_{s,i}$ [L T$^{-1}$] are thickness and saturated hydraulic conductivity of layer $i$, respectively, and $L = \sum z_i$ is the thickness of the layers above the water table. We compute the flux every 5 min according to the measured water levels in observation well OA. Note Eq. (1) provides temporal changes in flux because the properties (thickness and $K_s$) of the equivalent layer above the water table changes in time (Fig. 2a).

### 2.5.2 Analytical steady-state seepage model

The second analytical model assumes seepage flow through a perched water surface together with ponding depth data. We consider steady-state seepage through the topmost low-permeability SCL layer (Fig. 2a, 4–6 m depth), and that both this layer as well as the sand layer above it (0–4 m) are saturated under ponding, justified by the disparate hydraulic conductivities of the sand (high) and SCL (low) layers. By the same reasoning, the sand layer below the restrictive SCL layer remains unsaturated, maintaining steady-state flow (Fig. 2b). With the above, the 1D steady-state flux through the saturated layers $q_s$ can be described using Darcy's law as

$$q_s = \overline{K_s}(d + L - \psi^*)/L \tag{2}$$

where $d$ [L] is the ponding depth (measured every 5 min), $L$ [L] is the thickness of the saturated layers (sand and SCL) and $\psi^*$ [L] is the matric pressure head at the bottom of the saturated layers. The equivalent conductivity $\overline{K_s}$ for layers 1 to 3 (Table 1) was computed by Eq. (1). Assuming gravitational flow (unit gradient) in the unsaturated layer below the saturated layers (Zaslavsky, 1964), $\psi^*$ is estimated from $q_{un} = K(\psi^*)$, where $K(\psi^*)$ is the function relating the unsaturated hydraulic conductivity to $\psi^*$. Mass balance implies the equality of fluxes in the saturated ($q_s$, Eq. 2) and unsaturated ($q_{un}$) layers, providing $\overline{K_s}(d + L - \psi^*)/L = K(\psi^*)$. Here, we solve iteratively for $\psi^*$ using the van Genuchten–Mualem model (van Genuchten, 1980; Mualem, 1976), which was employed by others to estimate stream-aquifer seepage (Brunner et al., 2009; Osman and Bruen, 2002). Alternatively, to a leading order, one could take the $\psi^*$ as the atmospheric pressure or the sediment's air entry value. The flux in Eq. (2) is mostly affected by the ponding depth ($d$); $\psi^*$ is relatively insensitive to $d$ (varying by ±4%), and $\overline{K_s}$ and $L$ are constants (unlike the lumped model where these parameters are time-dependent).

### 2.5.3 Numerical model

Infiltration through the unsaturated zone was simulated with the HYDRUS-1D software (version 4.16), a finite element code for 1D uniform water movement in a variably-saturated rigid porous media (for a detailed description of the governing equations see Šimůnek et al., 2009). HYDRUS-1D was recently used to evaluate recharge in natural settings (Assefa and Woodbury, 2013; Neto et al., 2016; Turkeltaub et al., 2015). In this work HYDRUS-1D was used to evaluate infiltration through the unsaturated zone and groundwater recharge using the 1D Richards equation, with negligible root water uptake (sink term).

The model domain includes 10 layers within 0–30.5 m depth, with 8 different compositions based on the sediment core samples, discretized into 1000 elements of thickness of 0.1–4 cm (average of 3 cm) varied according to the sediment type and location, and to the original groundwater level (Fig. 2c).

The van Genuchten–Mualem model (van Genuchten, 1980; Mualem, 1976) was used for the water retention curves and unsaturated hydraulic conductivity functions of the different sediments. The hydraulic parameters (Table 1) were calculated from the measured particle size distribution and bulk density using PTF (ROSETTA, Schaap et al., 2001), which is incorporated in HYDRUS-1D. An exception was made for the Kurkar rock (layers 8 and 10), for which we used the hydraulic parameters of material 5 (sand) since the Kurkar was crushed during drilling and its structure was destroyed. The saturated hydraulic conductivities ($K_s$) of wells OA and OB were estimated by single-well recovery tests interpretation (data not shown) as 3.6 and 7.5 m d$^{-1}$, respectively. While these values represent an effective value of several layers in which the well screen crosses rather than a specific layer, $K_s$ values from these recovery tests are in the range of the values obtained by ROSETTA for the sand layers. In addition, previous studies (Levi, 2015; Shapira, 2012) showed that the local sediments at the Israeli Coastal aquifer show reasonable fit (R2=0.69, n=53) between the measured (in the laboratory) and estimated saturated hydraulic conductivity calculated by ROSETTA. Nevertheless, in order to test our numerical model results obtained with ROSETTA, we also preform simulations with two different USDA textural-classes PTF (Carsel and Parrish, 1988; Tóth et al., 2015, Tables S1 and S2 in the Supplement).

Boundary conditions at the soil surface (top) were the ponding depth (monitored at its surface when filled) and no flow when the pond is empty, and groundwater level (measured in well OA) at the bottom. The variable-head boundary conditions were applied at fixed locations at the top and bottom of the domain, and were updated at a one-hour resolution. Variable-head boundary conditions were selected in both boundaries in order to capture the highly dynamic behavior of the system during January 2015 MAR event (rise of 2.2 and 17 m in ponding depth and groundwater level, respectively). The code output is the flow at the domain boundaries: infiltration flux at the pond surface (top) and groundwater recharge (bottom). The recharge flux at the fluctuating groundwater table is similar to that at the bottom of the domain, because HYDRUS-1D neglects the storage term in the variably-saturated flow equation. Initial water content profile was obtained by field data (unsaturated at depth of 0–4 m and saturated below the water table, at 24.4 m) and by running a simulation for 88 days (September to December 2014) before the beginning of MAR that incorporates daily precipitation and evaporation.

## 3 Results

### 3.1 Monitoring pond and groundwater

The three pressure loggers monitored the local ponding depth during January 2015 MAR event, showing the filling period as water flows from the south to the north part of the pond (Fig. 3). A uniform water level was reached after the whole pond surface was covered with water and the three pressure loggers measured similar levels (since 3 January 2015), with a constant difference between the pressure loggers that represent ponding depth difference due to a shallower pond surface (topography) at the northern part. Next, water ponding depth increases sharply and reaches a maximum of 2.2 m (11 January 2015). At this stage a small dam was opened at the north part of the pond allowing water to flow freely to the connecting channel and to the northern infiltration ponds (Fig. 1b) and the ponding depth was stabilized at 1.8 m for 17 days. During this period, on 21 January 2015, a flooded area of 10.7 ha was mapped with a GPS device to obtain the ponded area (Fig. 1c). After 31 days, inlet discharge was stopped and the water ponding depth decreased; finally on 2 February 2015 the pond drained completely.

The pressure loggers in the observation wells captured a substantial rise of 17 m in groundwater level after 1 month of continuous MAR during January 2015 (Fig. 4a). Note that this rise represents the local conditions beneath the pond, while the influence on regional groundwater levels is damped farther away from the pond (e.g., a well, located 600 m to the north of the pond margins showed a maximal groundwater-level rise of 4 m; data not shown). The lag in groundwater rise (Fig. 3) is due to the infiltration process through the unsaturated zone, which takes around 3 days until water reaches the water table, since the beginning of discharge on 29 December 2014. However, groundwater drops almost immediately (5 hours) after inlet discharge was stopped and pond water level declines. This fast response implies on fully connected flow (Brunner et al., 2009) between the high level groundwater and the pond. Note that the relatively sharp decline in groundwater level during the first month after discharge was ended is followed by a gradual decline in the following months (Fig. 4a).

The EC monitoring of groundwater at the observation wells is shown in Fig. 4a to emphasize the response of groundwater to MAR with DSW (~0.2 mS cm$^{-1}$). Before the beginning of recharge with DSW on 29 December 2014, a 4 hours recharge with water from Lake Kinneret was applied on 25 December 2014. The higher EC of Lake Kinneret water (~1.1 mS cm$^{-1}$) compared to DSW is related with the high EC readings (up to 0.5 mS cm$^{-1}$) in the observation wells, which decreases as DSW reach the groundwater and finally the groundwater EC stabilized on 0.26 mS cm$^{-1}$.

Volumetric water content (WC) measurements in the vadose zone also capture the January 2015 MAR event showing constant WC during most of the ponding period and also later during the dry period. Changes in WC are most notable during wetting and drying of the upper sand layer at the beginning and end of infiltration, respectively (Fig. 4b). The bulk EC at depth of 3 m is increasing at the beginning of the MAR event and then decreases and stabilizes on 0.08 mS cm$^{-1}$, showing a similar trend as recorded in the observation wells and discussed above. The bulk EC (which is a function of the soil WC) decreases further to 0.01 mS cm$^{-1}$ when the soil drains and dries after the end of MAR.

### 3.2 Infiltration rates

#### 3.2.1 Pond scale

Pond infiltration rates show a general decrease during the January 2015 MAR event (Fig. 5a). Infiltration rates of $9\pm2$ (25 December 2014) and $2.9\pm0.4$ m d$^{-1}$ (30 December 2014) were calculated before the whole pond was flooded and represents an average infiltration rate of the temporal water body, which might be biased due to surface flow. Average pond infiltration rates of $0.84\pm0.02$, $0.72\pm0.08$ and $0.36\pm0.01$ m d$^{-1}$ were calculated after the whole pond was full. Multi-year average pan-evaporation for this area in January is 0.0023 m d$^{-1}$ (Israel Meteorological Service, 2016), which is 2 orders of magnitude smaller than the lowest infiltration rate measured for the MAR activity, hence evaporation losses are considered negligible hereafter.

#### 3.2.2 Single-ring scale

Results of the single-ring infiltration tests under the different conditions show some degree of spatial and temporal variability of infiltration rates (Fig. 5b). Spatial variability was evaluated from differences in rates in different locations, and was found to be moderate (coefficient of variation, CV=0.27) and high (CV=0.77) for dry and early ponding conditions, respectively. Infiltration rates during late ponding were measured when pond water level decreased and the infiltration rings were gradually exposed above the water line (from north to south). This process took 5 days and therefore the variability in late ponding infiltration rates can be considered both temporal and spatial. Nevertheless, the average single-rings infiltration rate of 10.8 (standard deviation, SD=2.9), 2.3 (SD=1.8) and 0.4 (SD=0.15) m d$^{-1}$ for dry conditions, early and late ponding, respectively, shows similar rates to the whole-pond rates results (Fig. 5a), indicating that the measured infiltration rates of the single rings were spatially representative.

#### 3.2.3 Wetting-front propagation

Sharp wetting and drying (drainage) fronts, typical for sandy and coarse sediments, were observed at the beginning and end of infiltration, respectively (Fig. 5c, d). The estimated infiltration rates between couples of soil sensors (i.e., between 0.3 and 0.5 m, 0.5 and 1 m, etc.) changes with time: the infiltration rate generally decreases as the wetting front advance deeper (Fig. 5e), as expected from theory (Assouline, 2013) and the impact of the SCL layer, while the drying front shows a more complex trend (Fig. 5f). In order to compare with infiltration rate measurements by other methods, the average infiltration and drainage rate of the top sand layer was estimated between the soil sensors in 0.3 and 4 m depth, as 12.8 and 0.16 m d$^{-1}$ for dry and post ponding conditions, respectively. The lower drainage rate of 0.16 m d$^{-1}$ compared to the average surface infiltration rate of 0.4 m d$^{-1}$ during late ponding (Fig. 5a, b), is expected due to the upward capillary tension exerted during internal drainage whereas this tension is absent during ponding.

### 3.3 Recharge models

#### 3.3.1 Simplified analytical models

The lumped and seepage models provide cumulative recharge of 20.2 and 16.4 m (2.2·10⁶ and 1.8·10⁶ m³ when multiplying by the active flood area), respectively. Both models assume constant flux along the profile at each time step, which means similar rates of groundwater recharge and surface infiltration. Average infiltration/recharge rates are 0.57 (SD=0.10) and 0.46 (SD=0.04) m d⁻¹ for the lumped and seepage models, respectively. These rates are in the range of the whole-pond measured infiltration rates (Fig. 5a).

#### 3.3.2 Numerical model

To capture temporal variations in drainage and groundwater recharge, we simulated the January 2015 MAR event from 25 December 2014 to 5 October 2015. We calibrated the numerical model to the whole-pond infiltration rate data (Fig. 5a) in order to generalize the local sediment profile into a whole-pond representative profile. Only saturated hydraulic conductivities of the top SCL section were modified during the calibration (4–6 m, layers 2 and 3, calibrated $K_s$= 0.38 m d⁻¹ for both layers).

The model calibration shows good fit for 90% of the infiltration period (4–31 January 2015) with a relative root mean square error of 4.8% (Fig. 6a). Relatively poor fit between the model and the whole pond infiltration data were obtained for the first two observations points (25 and 30 December 2014). These two observations were measured at early stages, when the pond was partly filled, which may overestimate infiltration rates due to surface flow. Checking the calibrated model against WC data (from the vadose zone monitoring system at 2 m depth) shows reasonable validation as the model was calibrated against whole-pond data, while the WC represents point-specific data. In terms of wetting/drying front, the model underestimates the arrival time of the front. A better fit to the WC data was achieved during the calibration process using the built-in HYDRUS-1D inverse modeling, by fitting the van Genuchten–Mualem parameters α, n and Ks (Fig. 6b). However, we decided not to use these calibration results (only calibrating Ks), as they underestimate the whole-pond infiltration rates, and hence the cumulative infiltration. Evaluating the calibrated model with our latest available MAR data (1.3·10⁶ m³, recharged during 12 January to 7 February 2016) shows good validation of the whole-pond infiltration rates with a relative root mean square error of 11.4% (Fig. 6c).

Testing the numerical model with different PTF shows an expected variation in the results of infiltration and recharge rates. These variations clearly demonstrate the need for calibration when using PTF to estimate deep vadose zone hydraulic parameters (Zhang et al., 2016). In this case, the simulation with the PTF of Tóth et al. (2015) was closest to the calibrated simulation results (Fig. S1 in the Supplement).

Our numerical simulation results highlight the transient nature of groundwater recharge. At the end of the January 2015 MAR event when the pond was empty (2 February 2015), the estimated total groundwater recharge was 17.1 m, vs. 22.4 m of surface infiltration (Fig. 7). That is, during ponding ~75% of the infiltrated water has reached groundwater, while the

remainder ~25% is retained in the newly-saturated zone between the pre-MAR water table (24.5 m below surface) and the gradually decreasing post-MAR water table. Out of these 'residual water', more than half reach the pre-MAR water table after ~1 month (~90% of the total infiltrated water), with the remainder arriving only after ~6 months, as can be seen from the change in the slope of the groundwater recharge curve (Fig. 7, red line).

For a flooded pond surface of 10.7 ha the total surface infiltration gives roughly a total water volume of $2.4 \cdot 10^6$ m$^3$ that was discharged to the pond. This is in a good agreement with $2.45 \cdot 10^6$ m$^3$ that was reported by Mekorot that supplied the water. Comparison of the estimated recharge by the simplified models and the numerical model is shown in Fig. 7. Both simplified model underestimate the total infiltration, but the lumped model is closer (20.2 m) than the seepage model (16.4 m) to the numerical model (22.4 m).

**4 Discussion**

 **4.1 Spatial and temporal variability of infiltration rates**

Spatial infiltration variability depends on the soil type and structure and its spatial distribution in the pond. Single-ring dry infiltration rates showed significant correlation with bulk density (r=-0.57, p=0.003) sampled at the soil surface (5 cm deep), but for the same samples no significant correlation was found with water content or with clay and silt fraction. Also for the

60 cm-deep samples, no significant correlation was found with clay and silt fraction. Because the upper soil in all the infiltration rings is classified as sand (at least 97% sand) it is likely that very minor differences in the soil structure and particle size distribution are responsible for the difference between rings under dry infiltration rates (Fig. 5b). These minor changes are probably below the resolution of the particle size analysis that was conducted on all the sediment samples in this study. During the early ponding infiltration rate measurements (starting 4 hours after ponding started), the wetting front was

advancing further downward into the profile after it passed the top sandy layer, as evident by the soil sensors readings (Fig. 5c). At this stage, the spatial variability of infiltration rates is probably controlled by the lithology of the deep layered soil profile.

Temporal infiltration variability is evident from the single-ring tests as infiltration rates decrease from 6–16 m d$^{-1}$ before MAR (dry conditions) to 0.1–0.7 m d$^{-1}$ at the end of the MAR operation (late ponding, Fig. 5b). This temporal variability is

similar to the variability from pond scale (Fig. 5a) and vadose zone infiltration rates (Fig. 5c–f). The main reason for this 2 orders of magnitude decrease of infiltration rate is the sharp contrast between the hydraulic conductivity of the top sand layer ($Ks = 13.5$ m d$^{-1}$) and the SCL layer underneath ($Ks = 0.07$ m d$^{-1}$, Table 1). This layer has the lowest hydraulic conductivity along the unsaturated profile, so it serves as the limiting layer for pond infiltration. Thus, the fast infiltration rate at early ponding continues as long as water flows to the north part of the pond during the pond-filling process and simultaneously the

wetting front has not reached the SCL layer.

High spatial and temporal variability of infiltration rates, measured with thermal and pressure probes, was reported by Racz et al. (2012) during several months of MAR to an infiltration pond with an area of 3 ha. They postulated that small

differences in the percentage of fine material in the relatively homogenous shallow soil, clogging of the pond surface and deeper unsaturated zone processes can explain this variability. Mawer et al. (2016) used fiber optic distributed temperature sensing to monitor infiltration rates with high spatial resolution during MAR to an infiltration pond. They concluded that 80% of the recharged water infiltrated through the most permeable 50% surface area of the pond which was explained by

5 heterogeneous clogging. In our study the relatively deep unsaturated zone sampling and infiltration rate data shows that the spatial and temporal variability of infiltration rates is suppressed (and controlled) by the low-permeability layers. Probably for the same reason, together with the high-quality source water (DSW), there was no field evidence in our study for clogging of the top sand layer.

## 4.2 Clogging

Clogging of the infiltration surface is the major operational concern in most MAR systems (Bouwer, 2002; Martin, 2013). The extent of clogging during MAR with DSW is questionable due to the low turbidity, organic matter and total dissolved solids (TDS) of the source water (in this case, the DSW turbidity ~0.2 NTU and TDS ~120 mg L$^{-1}$). Vandenbohede et al. (2009b) reported on pond clogging during MAR with reverse-osmosis desalinated wastewater (TDS = 50 mg L$^{-1}$). It was explained by the accumulation of algae on the pond bottom, but the authors stated that "further research is needed to explain

the reasons for the clogging". In laboratory experiments, Mukhopadhyay et al. (2004) reported on permeability reduction following injection of filtered (<0.5µm) DSW into cores initially saturated with groundwater. The authors explained this reduction by clogging with fines originating from dissolution of carbonate and gypsiferous matrix, commenting that further research is needed. In this study we did not find evidence for biological clogging, while dissolution of carbonate is a minor concern as the DSW that was used here is enriched with calcium during post-treatment of the desalination process, and

therefore the DSW is saturated with respect to calcium-carbonate (Ronen-Eliraz et al., 2017).

To further examine the impact of DSW on clogging, we performed preliminary infiltration column experiments in the laboratory with DSW and sand taken from the pond surface (top 0.4 m). Results showed a reduction by a factor of 1.5 compared to the initial infiltration rate, probably due to compaction-clogging (see section S2 in the Supplement). Similar results were obtained by Lado and Ben-Hur (2010) in a column experiment with sandy soil leached with reverse-osmosis

effluent. They suggested that the relatively large average pore size in the sandy soil prevented pore clogging and $K_s$ reduction. These findings support our field-scale assumption that infiltration-rate reduction due to clogging processes at the top sand layer is absent or negligible, compared to the impact of the low-permeability layers. This is also supported by the numerical model validation (Fig. 6c) as infiltration dynamics is captured nicely without the need to incorporate a clogging parameter in the numerical model.

It is worth noting that infiltration of low-salinity water into natural settings may cause clogging due to clay swelling, dispersion and colloidal release and deposition, which can lead to $K_s$ reduction of up to 2 orders of magnitude (Blume et al., 2002; Lado and Ben-Hur, 2010; Mohan et al., 1993; Shainberg and Letey, 1984). The practical importance of these clogging

mechanisms during MAR with DSW is unclear. Further research is needed in order to determine the long term impact of MAR with DSW on field-scale clogging at the Menashe site.

## 4.3 Groundwater recharge

Clearly, the simplified models cannot capture groundwater recharge dynamics as the numerical model does, but they can
serve as a first approximation for recharge when no other data is available or as complementary recharge estimation when other methods are used. The lumped model suffers from practical and theoretical limitations compared to the seepage model. The need for drilling a monitoring well inside (or very close to) the infiltration pond and in addition continuously monitoring groundwater level is the main operational limitation. Moreover, there is no field evident that supports the assumption of a saturated profile with a unit gradient along the heterogeneous sediment profile. Yet, a main advantage of the lumped model
is its ability to predict groundwater recharge using only pedotransfer-based $K_s$ values when water table data is available. Noticeably, even when water table data is unavailable, using the equivalent hydraulic conductivity ($\overline{K_s}$) of the sediments above the regional groundwater level provides an excellent estimate of the total cumulative recharge (in this study, $\overline{K_s}$ = 0.73 m d-1, gives a total of 22.6 m during 31 days of MAR). The seepage model is practically simpler, as it does not require deep drilling (more relevant to settings with thick unsaturated zone) and only continuous monitoring of the pond water level is
needed. While the seepage model requires the occurrence of a shallow low-permeability layer beneath the pond, it is not considered a major limitation because a clogging layer is usually found in most MAR systems (Bouwer, 2002).

The unit-gradient assumption in the simplified models was tested using the results of the calibrated numerical model. Checking the hydraulic gradients as calculated from the calibrated numerical model, for the lumped model (between the pond surface and the groundwater table) and for the seepage model (at 6 m depth, below the upper SCL layer) shows that the
unit-gradient assumption is not always valid (Fig. 8a). This is due to the significant water table rise, the layered sediment profile and the variably saturated conditions. These factors, together with the lack of calibration of the simplified models, provide a possible explanation for the differences between the simplified models and the calibrated numerical model.

The calibrated 1D numerical model is a more complex tool compared to the simplified models that were presented or to other approximated methods, yet it is still simpler compared to 2D or 3D variably saturated models. Sumner et al. (1999) and
Morel-Seytoux (2000) discussed the validity of 1D flow along the unsaturated zone for estimating groundwater mounding during MAR. We tested our numerical model results using the analytical solution of Morel-Seytoux et al. (1990), which assumes 1D vertical infiltration along the unsaturated zone and radial flow along the saturated zone. The calculated groundwater level below the pond (Eq. (27) in Morel-Seytoux et al., 1990) using the infiltration and recharge rates results from the calibrated 1D numerical model, shows a reasonable fit with the observed groundwater levels, supporting our
assumption that flow along the unsaturated zone is mainly vertical (Fig. 8b). The differences between the calculated and observed groundwater levels can be attributed to the analytical model assumptions and to errors associated with the estimated model parameters (0.25, 185 and 70 m for specific yield, equivalent pond radius and saturated aquifer thickness, respectively).

The main advantage of our 1D numerical model is its ability to capture infiltration and recharge dynamics of the MAR system based on only one representative sediment profile. The obvious main drawback of the 1D model is its inability to capture lateral flows, both at the unsaturated and saturated zones. This limitation, to some extent, is compensated by the use of data-based variable-head boundary conditions which were employed to better estimate surface infiltration and groundwater recharge (for comparison, applying a constant-head lower boundary condition as an alternative will overestimate recharge). However, when using these boundary conditions the model is inadequate for predicting water table or ponding depth evolution during future MAR events. This limitation can be overcome by changing the model boundaries as discussed, for example, by Neto et al. (2016).

Predicting the recharge dynamics during MAR by numerical simulations is a valuable tool for planning successive MAR events, and as an input for regional groundwater models. Groundwater recharge is governed by the boundaries of the system and the unsaturated zone hydraulic properties. In MAR sites with unsaturated zone of intermediate depth (normally ~25 m at the Menashe MAR site), the water storage of the unsaturated zone affects infiltration and recharge dynamics. This is shown in Fig. 9 and can be divided into three stages: (1) high infiltration and low recharge rates during the saturation process of the vadose zone; (2) full water capacity (or close to) is attained; infiltration and recharge rates are similar and finally converge and decreases due to groundwater level rise (hydraulic gradient decreases); and (3) end of ponding (infiltration ends); recharge rate and water storage decrease further during drainage of the vadose zone. In MAR sites with shallow or no unsaturated zone (e.g., Vandenbohede et al., 2008), stages 1 and 3 are minor (if any) and the system will persist at stage 2 during the MAR operation. On the other hand, MAR sites with very deep unsaturated zone (e.g., Flint et al., 2012) may skip stage 2, crossing from stage 1 to 3 without reaching the potential water storage. The optimal extent of each stage during MAR operation is site-specific, depending on the MAR site requirements and constraints.

## 5 Summary and Conclusions

Groundwater level under a sandy infiltration pond in the Israeli Coastal Aquifer rose by 17 m during 1 month of continues MAR with surplus desalinated seawater. Measured infiltration rates were relatively uniform spatially, however highly variable in time: during continuous discharge of $2.45 \cdot 10^6$ m$^3$, rates decreased by almost 2 orders of magnitude. This reduction can be explained solely by the lithology of the unsaturated zone that includes relatively low-permeability sediments, whereas clogging processes at pond-surface are negated by the high-quality desalinated seawater or negligible compared to the low-permeability layers. While sediment sampling and analysis is a routine procedure in hydrology science, we emphasize its crucial role in MAR projects. Careful consideration of the hydrological properties of the deep unsaturated zone is needed in order to quantify the contribution of the local sediments to infiltration rate dynamics, compared to the contribution of the MAR-related clogging layer. To date, literature on clogging during MAR with desalinated seawater is limited and the extent of field-scale clogging is unclear and probably site-specific. For this reason, the long term impact of MAR with desalinated seawater on clogging processes at the Menashe site should be addressed in future studies.

Groundwater recharge was estimated by analytical and numerical models that include ponding and groundwater head data. The simple analytical models can estimate reasonably well cumulative groundwater recharge using field data, but predicting the late recharge after pond infiltration terminates, requires a detailed unsaturated flow model. A one-dimensional numerical model with a whole-pond representative soil profile can capture groundwater recharge dynamics, especially when it constrained by measured variable-head boundary conditions. Validation of our numerical model in an independent MAR event shows the model robustness. The dynamic groundwater recharge described by the numerical model is useful for future MAR operation planning, and also as input for regional-scale groundwater modeling.

## Acknowledgements

The research leading to these results received funding from the European Union Seventh Framework Program (FP7/2007-2013) under grant agreement no. 619120 (Demonstrating Managed Aquifer Recharge as a Solution to Water Scarcity and Drought – MARSOL). We thank Amos Russak and Raz Amir for their technical assistance and Eline Futerman-Hartog for conducting the dry infiltration experiments.

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

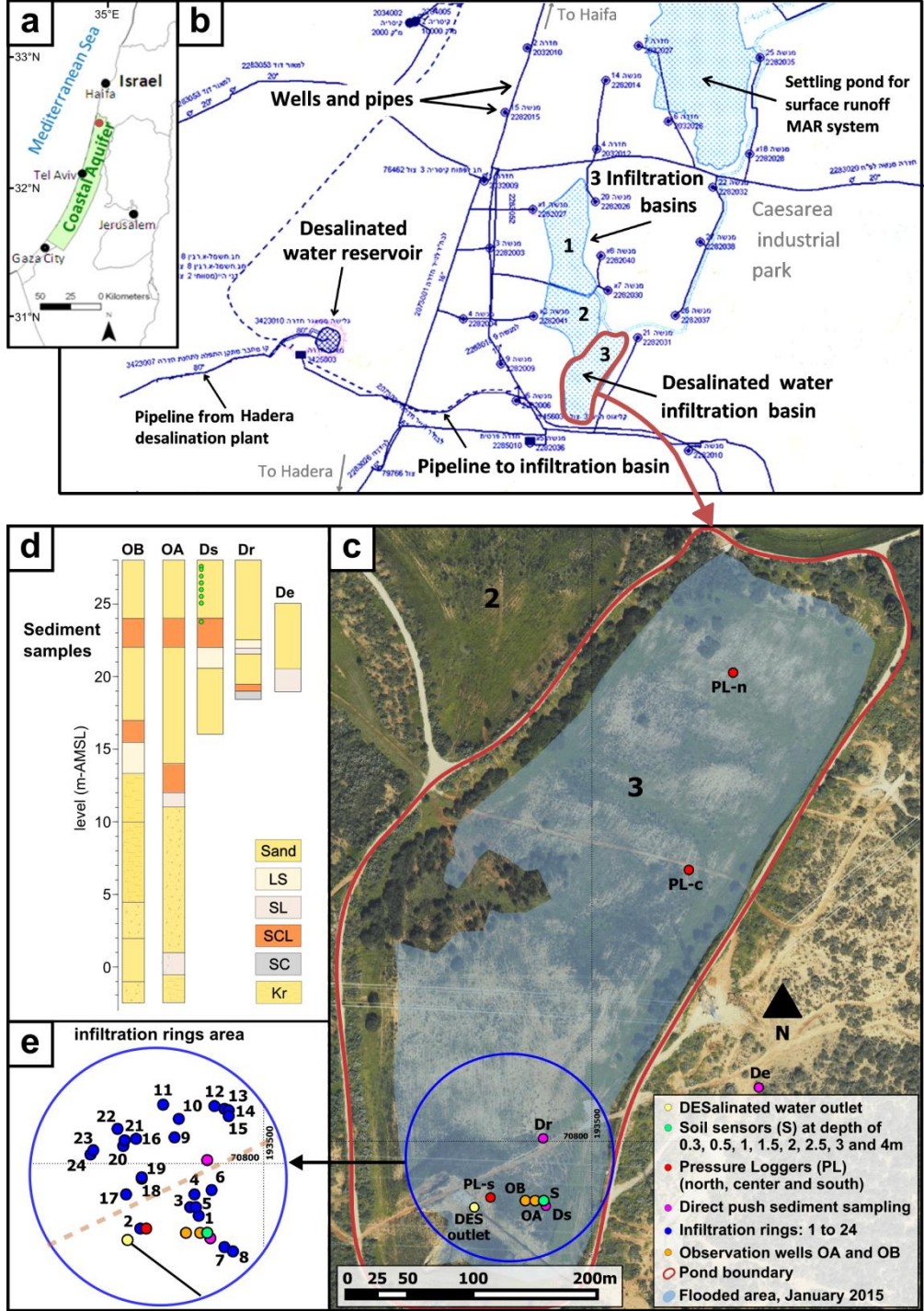

**Figure 1. (a)** Location of the Israeli Coastal Aquifer and the Menashe site (red circle). **(b)** Menashe MAR site. **(c)** Southern infiltration pond with the monitoring system. **(d)** Observation wells (OA and OB) and direct-push (Ds, Dr and De) sediment profiles. The soil sensors are shown schematically on profile Ds. LS – loamy sand; SL – sandy loam; SCL – sandy clay loam; SC – sandy clay; and Kr – Kurkar. **(e)** Infiltration-rings locations (1–24).

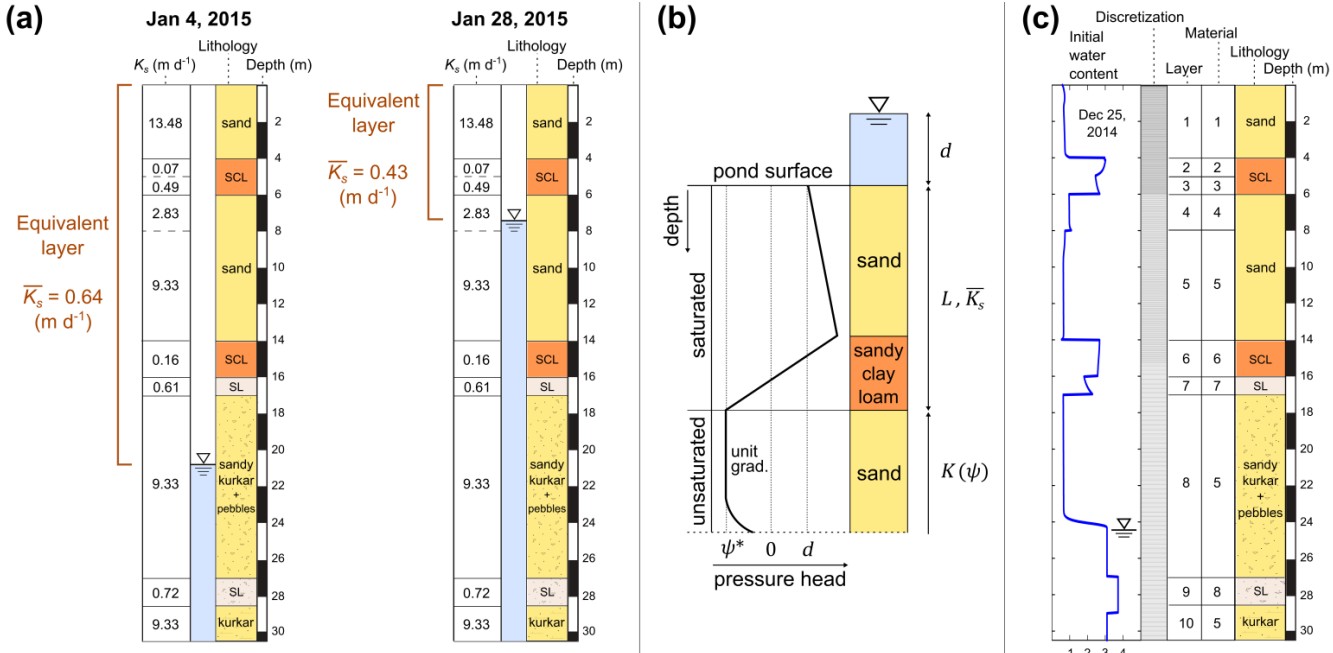

**Figure 2. Recharge models: (a) Estimation of fluxes at early (right) and late (left) stage of the January 2015 MAR event in the lumped model. $\overline{K_s}$ is the equivalent vertical hydraulic conductivity. (b) The saturated portion with thickness $L$ and $\overline{K_s}$ is used to estimate flux in the seepage model. A schematic steady-state pressure head profile shows the transition from saturated to unsaturated conditions (after Zaslavsky, 1964). (c) The numerical model domain.**

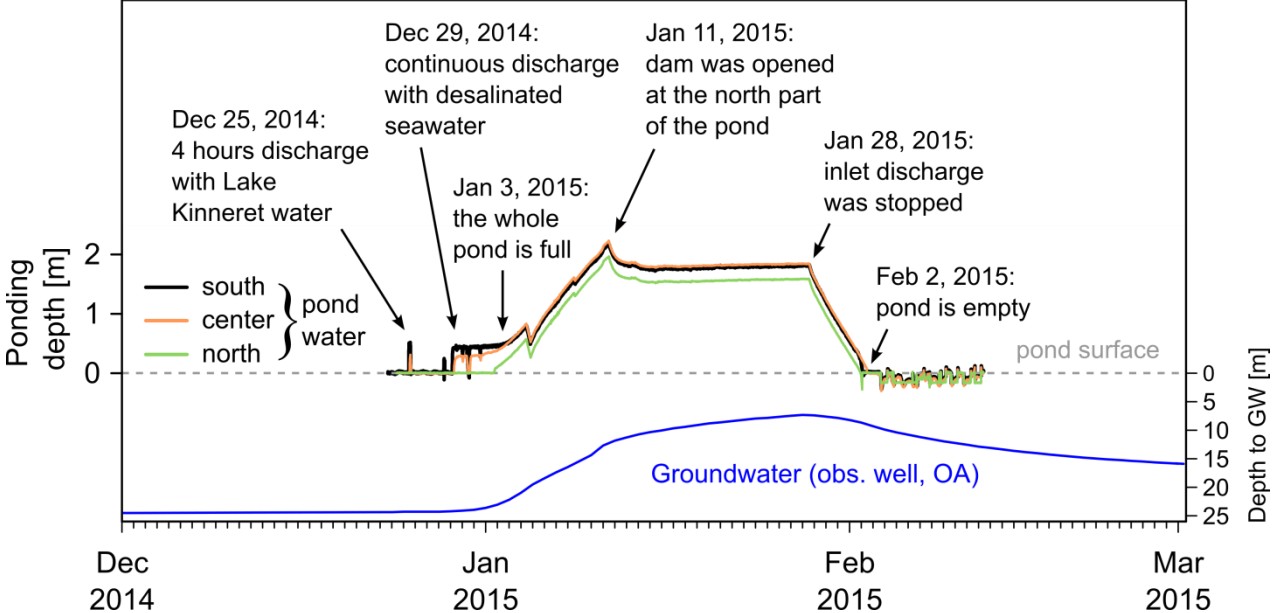

**Figure 3. Water ponding depth at the south, center and north locations inside the pond during the January 2015 MAR event. Observation well OA shows a sharp increase of the water table during MAR (17 m). Minor ticks on the X-axis are days.**

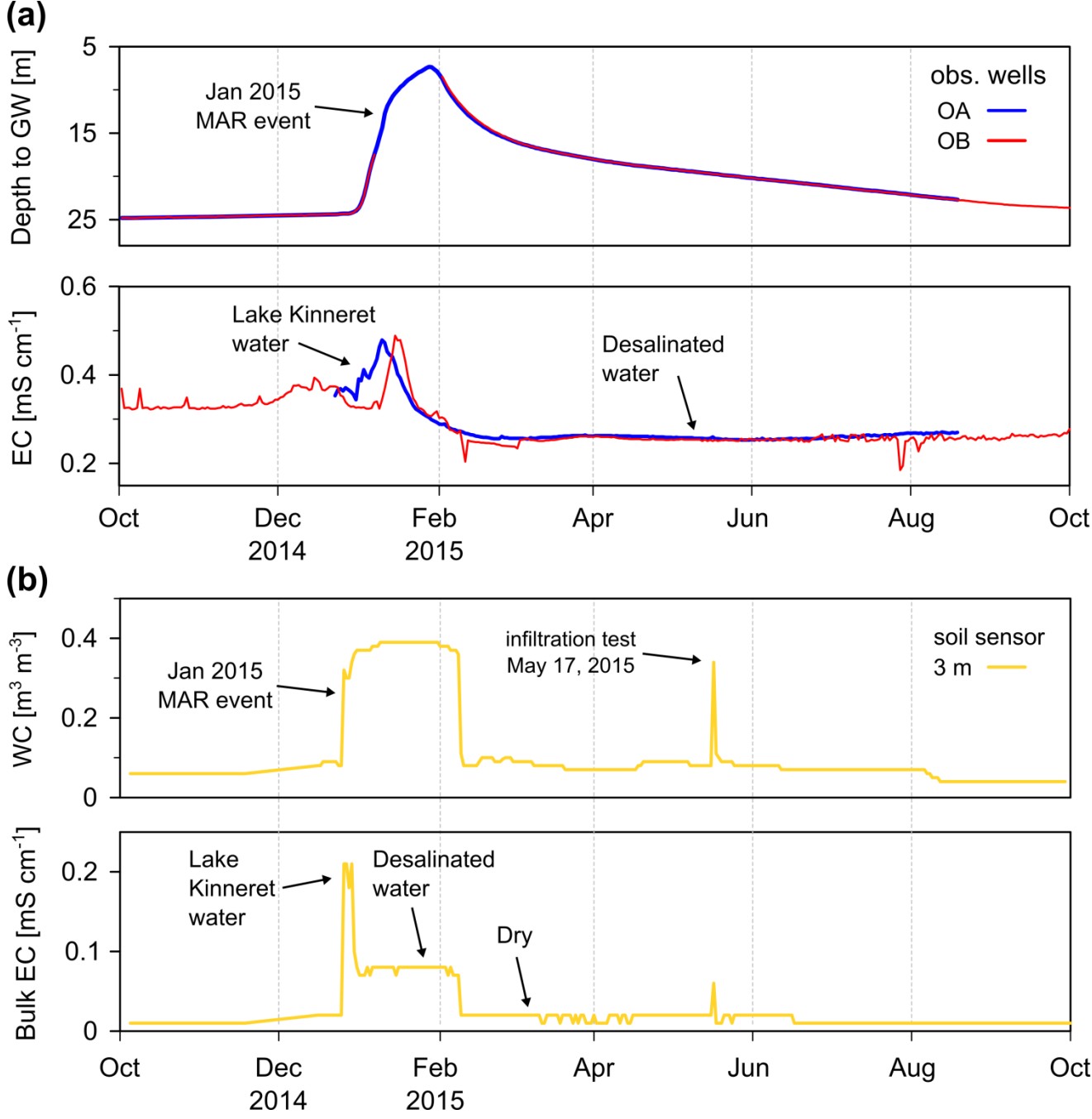

**Figure 4. (a)** Monitoring of groundwater level and EC in observation wells OA and OB during one year. **(b)** Volumetric water content (WC) and bulk EC at 3 m below ground surface. Note that the January 2015 MAR event and changes in water quality can be seen in both groundwater and vadose zone monitoring.

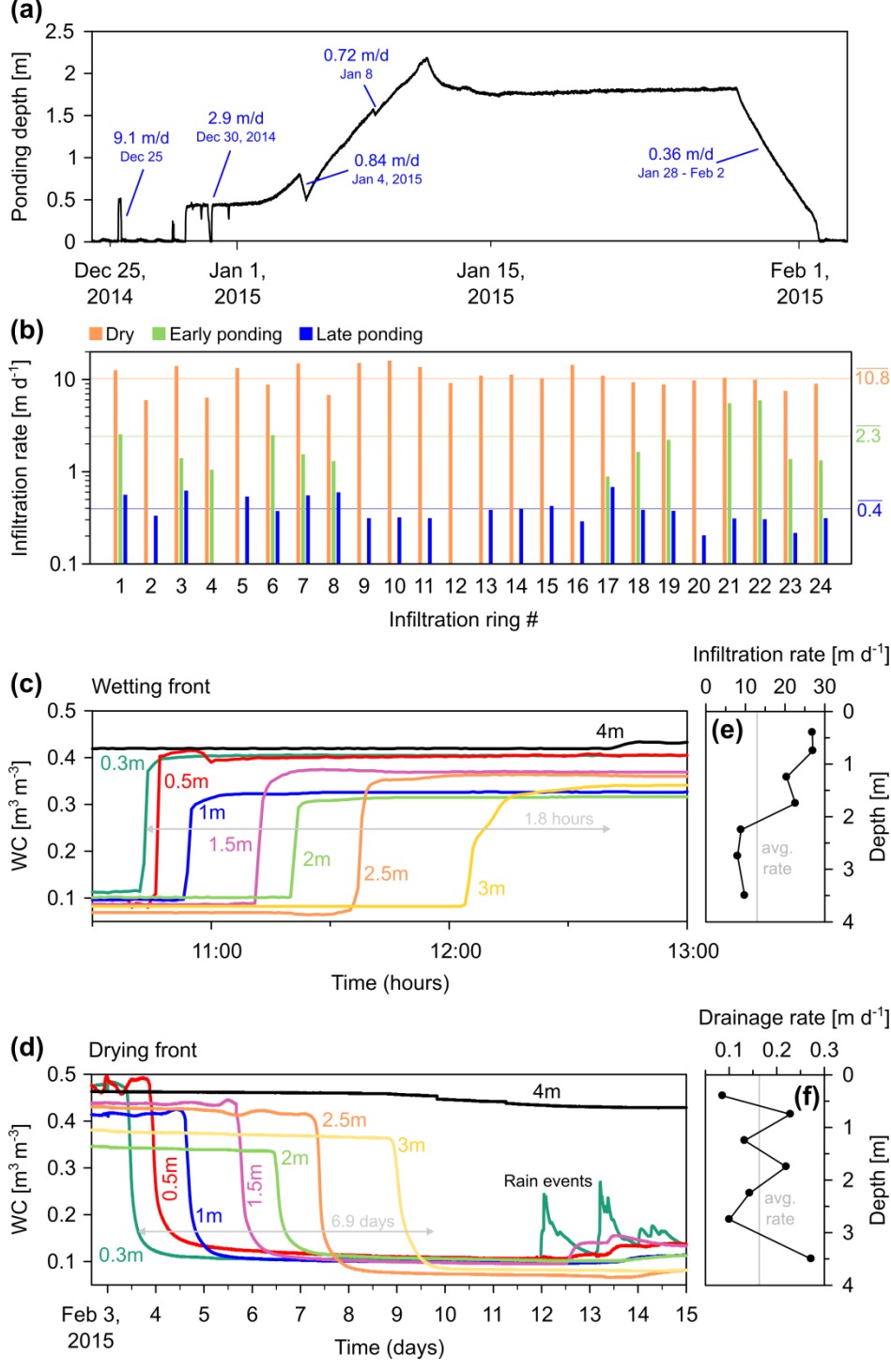

**Figure 5. Infiltration rate measurements at various scales and perspectives: (a) pond scale (10.7 ha); (b) Infiltration rings (0.1 m²), note the log-scale at the Y-axis; (c) wetting front propagation in the vadose zone at the beginning of infiltration; (d) drying front propagation in the vadose zone at the end of infiltration; (e) infiltration rates calculated from wetting and (f) drying front.**

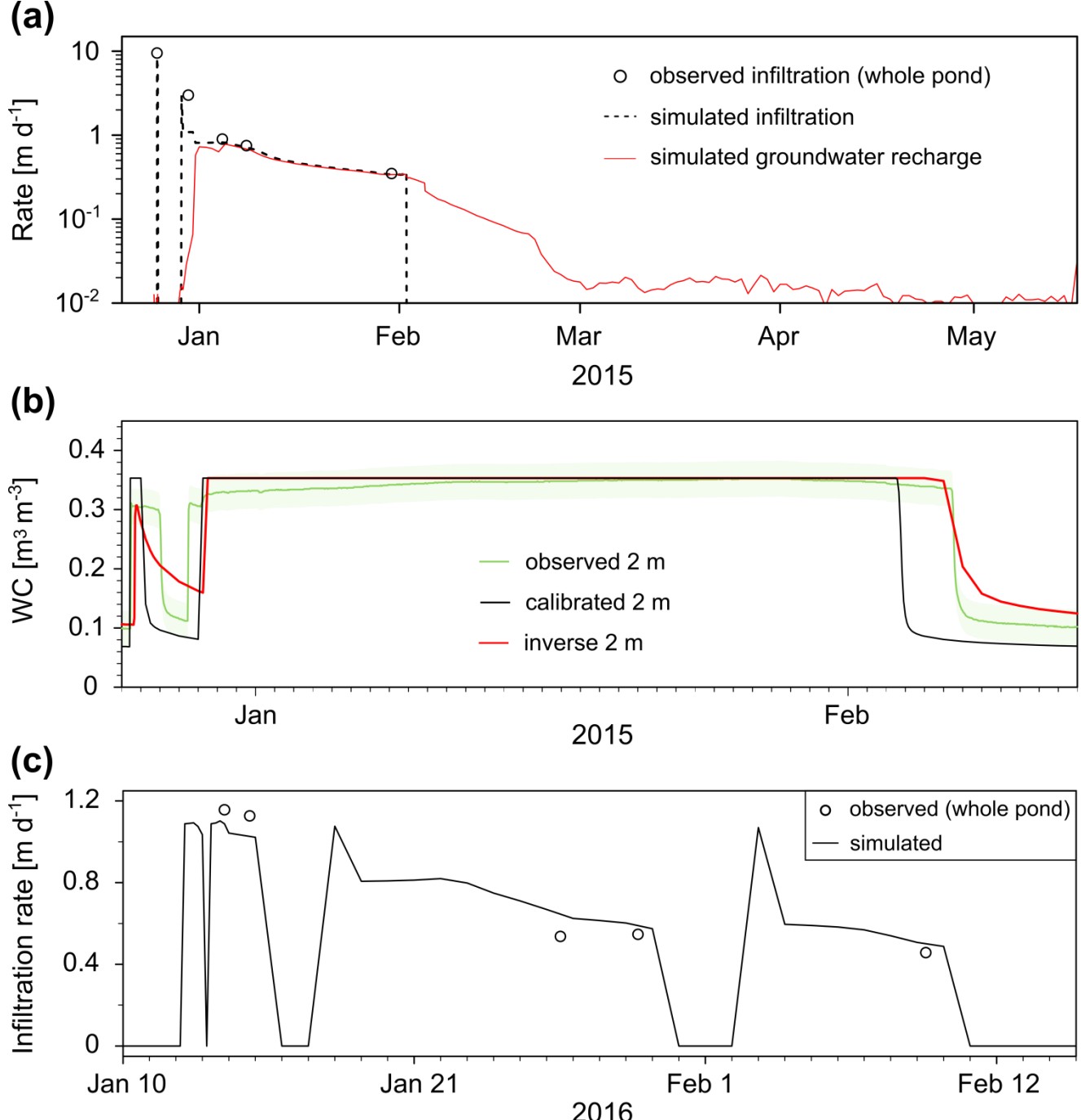

**Figure 6. (a)** Calibration of the infiltration rates at ground surface in the pond; note the log-scale at the Y-axis. The simulated groundwater recharge rate is also shown for comparison. Note the large difference between simulated infiltration and recharge rates at the beginning of the MAR event versus the identity in rates that is achieved after a few days, and the continuation of recharge after the end of the infiltration. **(b)** Validation fit of the calibrated and inverse models to volumetric water content (WC) data from depth of 2 m. **(c)** Validation fit of the calibrated model to January 2016 MAR event (1.3·10⁶ m³). Minor ticks on the X-axis are days in (b) and (c).

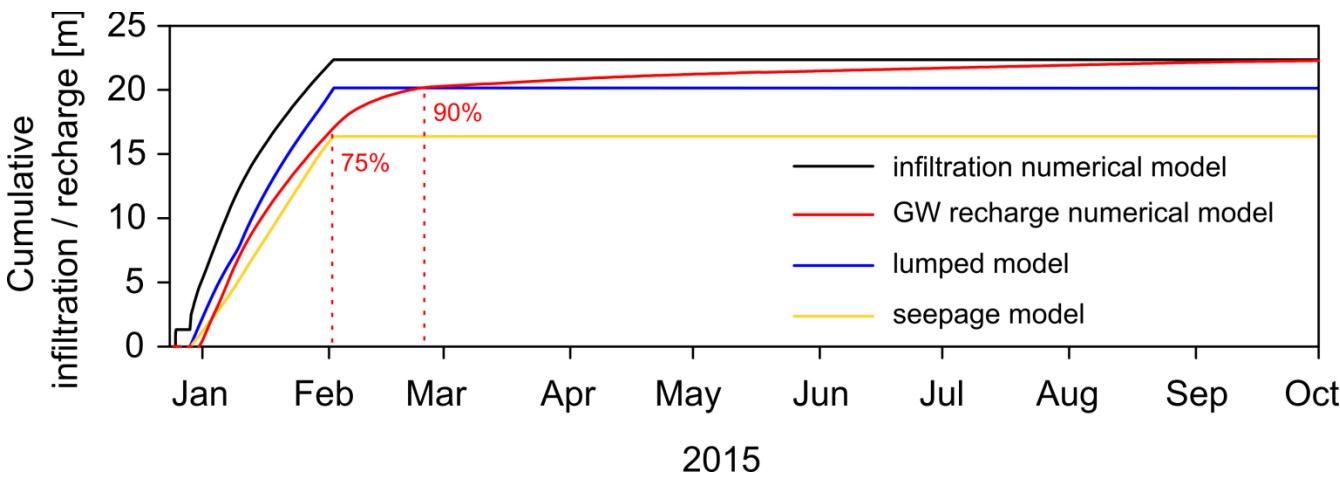

**Figure 7. Cumulative infiltration and recharge of the various models during 2015. According to the numerical model, most of the infiltrated water (~75%) reached the pre-MAR water table after 1 month (end of ponding), and ~90% reached after about two months since the onset of infiltration.**

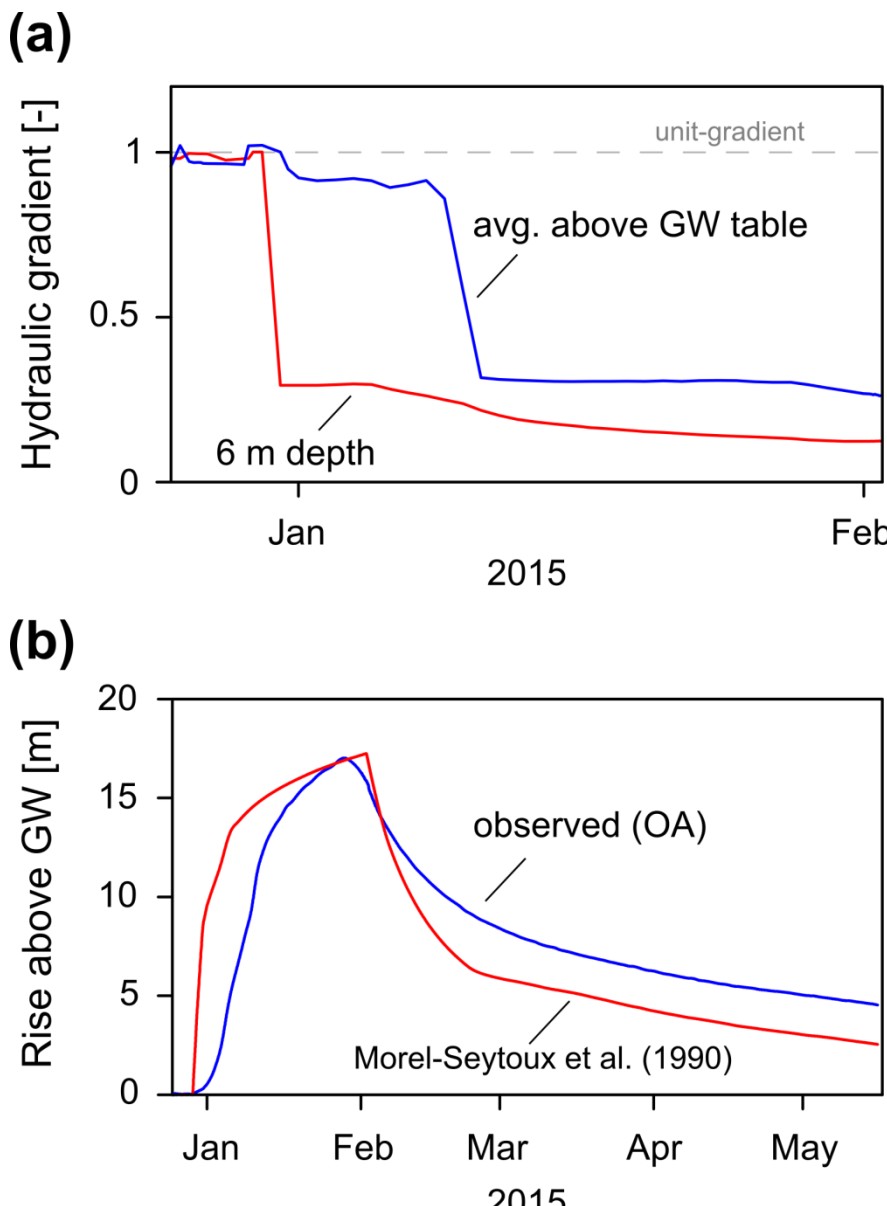

**Figure 8. Testing models assumptions. (a)** Unit-gradient assumption: the hydraulic gradients (HG) were calculated from the numerical model to test the HG of the lumped model (average HG above the fluctuating groundwater table) and of the seepage model (HG at 6 m depth). **(b)** 1D flow assumption along the variably-saturated zone: measured groundwater rise above the original groundwater table compared to the analytical solution of Morel-Seytoux et al. (1990) calculated using the infiltration and recharge fluxes of the numerical model.

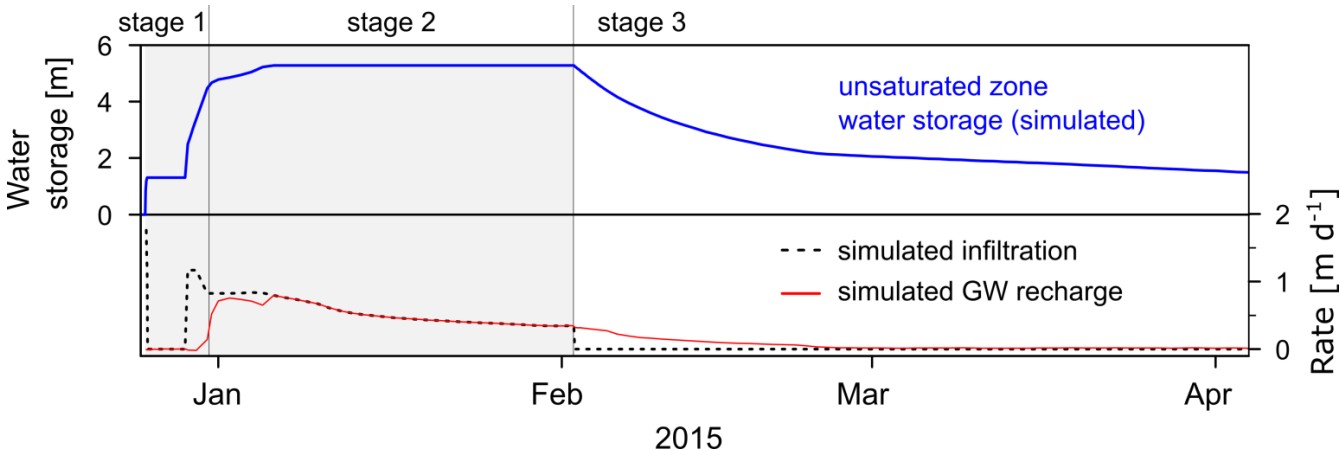

**Figure 9. Change in water storage in the unsaturated zone (i.e., above the original water table) during ponding (stage 1 and 2) and drainage (stage 3). Simulated rates of surface infiltration and groundwater recharge are also shown as a reference.**

**Table 1.** The material properties used for the layers in the analytical models and uncalibrated numerical model (Fig. 2): soil texture and van-Genuchten–Mualem hydraulic functions parameters: residual and saturated water contents, $\theta_r$ and $\theta_s$; fitting parameters α and n; and saturated hydraulic conductivity, $K_s$.

| Material | Soil texture | $\theta_r$ | $\theta_s$ | $\alpha$ (m$^{-1}$) | n | $K_s$ (m d$^{-1}$) |
|---|---|---|---|---|---|---|
| 1 | Sand | 0.053 | 0.353 | 3.03 | 4.52 | 13.48 |
| 2 | Sandy clay loam | 0.063 | 0.336 | 2.9 | 1.19 | 0.07 |
| 3 | Sandy clay loam | 0.068 | 0.387 | 2.42 | 1.52 | 0.49 |
| 4 | Fine Sand | 0.052 | 0.300 | 2.96 | 2.85 | 2.83 |
| 5 | Sand/Kurkar | 0.050 | 0.310 | 3.09 | 4.05 | 9.33 |
| 6 | Sandy clay loam | 0.068 | 0.369 | 2.46 | 1.32 | 0.16 |
| 7 | Sandy loam | 0.064 | 0.365 | 2.32 | 1.71 | 0.61 |
| 8 | Sandy loam | 0.056 | 0.372 | 2.9 | 1.66 | 0.72 |