# Peer review of "Monitoring and modeling infiltration-recharge dynamics of managed aquifer recharge with desalinated seawater"

_Hydrology and Earth System Sciences, 2016_

## Referee Comment (RC1) · Anonymous Referee #1 · 17 Apr 2017

This manuscript describes a field experiment conducted and numerical exercises used to evaluate infiltration-recharge dynamics of managed aquifer recharge with desalinated seawater. While their efforts on the field experiments are admirable and the topic is of great interest to the reader of Hydrology and earth System Sciences. I am not good for managed aquifer recharge with desalinated seawater. However, I have a few comments which hopefully might be helpful for further improvement of the paper.

I noticed that the authors' usage of calibration and validation. "The model calibration shows good fit for 90% of the infiltration period (4–31 January 2015) with a relative root mean square error of 4.8% (Fig. 6a)." There are only five points in Fig. 6a, and five points in Fig. 6c?

I am confused with "Only the saturated hydraulic conductivity of the top SCL was modified during calibration of the numerical model.? " Why "$\theta$r $\theta$s $\alpha$ (m-1) n " were not modified during calibration of the numerical model?

P11ïïjŇ"In the laboratory, infiltration column experiments with DSW and sand taken from the pond surface (top 0.4 m) showed a reduction by a factor of 1.5 compared to the initial infiltration rate due to compaction-clogging (data not shown)." I think this sentence is not closely related to the above passage.

---

## Referee Comment (RC2) · Anonymous Referee #2 · 10 May 2017

Monitoring and modeling infiltration-recharge dynamics of managed aquifer recharge with desalinated seawater by Ganot Y. et al.

Comments to authors: Based on the solid monitoring and modeling work, the authors are trying to explain the relationship between surface infiltration and groundwater recharge during MAR with desalinated seawater in one pond located in the coastal Israel. Currently, there is few researches done on MAR with DSW in the coastal areas of the world. They obtained a good dataset. However, the authors did not give good illustration on these data and find the main focus of the paper. More consideration as to how this study can inform others elsewhere in the world. I hope that the comments are useful to the authors in revising this study.

Specific comments:

Page 4 Line 8: "is" should be 'are".

2.4 section: Page 4: How to calculate infiltration rates for different conditions should be introduced here. For pond scale, the pond infiltration rates were calculated by linear-regression of ponding-depth declines due to breaks in inlet discharge during MAR event. For single-ring scale, the single-ring infiltration rates were calculated by linear-regression of pressure-head vs. time data. Why did you define that "Infiltration tests with relative errors higher than 15% were omitted from the analysis."? For vadose zone: Infiltration rates were estimated from the velocity of the fronts multiplied by the change in volumetric water content data. For the three aspects mentioned above, could you give the corresponding references or clear conceptual sketch map to illustrate this parameter? This will help readers better understand the infiltration rates under different conditions.

Page 4, Lines 29-31. As the numerical model used both the water table and ponding-depth data, what is purpose of the two analytical model under simplified condition?

Page 5, Lines 14. The impacts of the assumption of the unit gradient flow for steady state should be evaluated under the transient condition as the water table rise significantly during the recharge period.

Page 6, Line 32-33. It is not very clear to me how the variable head boundary was applied at the bottom boundary of the Hydrus-1D model and how simulate the transient behavior of the water table. Did you specify variable head at a fixed location and output the flux at this boundary? Please clarify this.

Page 10, Lines 12-14. Another reason why the numerical model seems more reliable than the simplified model may be that the numerical model was calibrated based on observations. I think some testing work of the assumptions (i.e., saturated condition under the pond in the) using the flux distribution from the HYDRUS may be useful for

the comparison of the simplified model and numerical model.

Page 11: 4.2 Clogging Line 21-24: Do you have the hydrochemical data to provide evidence for the dissolution of carbonate and gypsiferous matrix?

Page 12 Line 15-25: this part should be shifted to Methods section.

This manuscript estimated groundwater recharge using one dimensional (1D) analytical models and 1D numerical models. The reasons of using 1D models presented in the manuscript is not justified. It is better to present the results using 2D or 3D models, and compare the results 1D models.

As was indicated by Zhang et al (2016) that using soil pedotransfer functions directly may lead to biased estimation of dynamic soil moisture content, it is necessary to use site specific pedotransfer functions or inverse models to improve the soil hydraulic parameters used in both analytical models and numerical models. Assouline and Or (2013) also wrote that "The empirical-correlative approach at the basis of the PTF offers reasonable initial estimates for certain large-scale analyses (Romano, 2004). However, the limited physical basis for the estimates of WRC and applicability within the range of values used for the regression analysis, necessitate extra caution for their general application". The authors may use other pedotransfer functions to compare the results by using Rosetta presented in the manuscript.

Zhang, Y., Schaap, M.G., Guadagnini, A., Neuman, S.P., 2016. Inverse modeling of unsaturated flow using clusters of soil texture and pedotransfer functions. Water Resour. Res. 52, 1–14 Assouline, S., and D. Or (2013), Conceptual and parametric representation of soil hydraulic properties: A review, Vadose Zone J.,12, 1-20, doi:10.2136/vzj2013.07.0121. Romano, N. 2004. Spatial structure of PTF estimates. In: Y.A. Pachepsky and W.J. Rawls, editors, Development of pedotransfer functions in soil hydrology. Elsevier Science, New York. p. 295-319.

In the Conclusions section explain in more detail how your project helps us to

understand processes in these environments more broadly; the paper will have more impact if researchers from elsewhere in the world can see relevance to their studies and a paper in a major international journal such as HESS needs to have broad appeal.

Please also note the supplement to this comment:
http://www.hydrol-earth-syst-sci-discuss.net/hess-2016-613/hess-2016-613-RC2-supplement.pdf

---

## Author Comment (AC1) · 10 May 2017

We would like to thank the Anonymous Referee #1 for his review and comments. We provide below our detailed response to each comment.

Comment 1: I noticed that the authors' usage of calibration and validation. "The model calibration shows good fit for 90% of the infiltration period (4–31 January 2015) with a relative root mean square error of 4.8% (Fig. 6a)." There are only five points in Fig. 6a, and five points in Fig. 6c?

Response 1: The numerical model in this study was calibrated to whole-pond data in order to capture the infiltration dynamics of the whole-pond infiltration event (P9.L19).

Most models are calibrated against head data, but here we were fortunate to be able to calibrate the model to flux data (five points) which usually are unavailable at all. Other methods for estimating infiltration rates are point-specific (usually indirect) and therefore were not suitable for this purpose. The numerical model was calibrated (and validated) using whole-pond infiltration rates (fluxes) that were calculated by linear-regression of the 5-minutes-resolution ponding-depth data (P8.L9 and Fig. 5a). Two conditions must be met in order to calculate infiltration rates by this method: (1) ponding depth is declining solely due to infiltration (i.e., no other inlet\outlet source or surface flow) and (2) span of the descending ponding-depth data is sufficiently long (usually at least few hours) in order to obtain regression with low-error slope (which is a good esti-mate of the integrated pond-infiltration-rate). Condition (1) was the limiting factor during the operative MAR events and therefore we obtained only five observation points for each calibration and validation. Yet, we emphasize that each observation point was calculated from a large number (tens to hundreds) of ponding-depth data measure-ments. It should also be acknowledged, that unlike most field-scale flow models, the top and bottom boundary conditions of this model were continuously measured head values (thousands of measurements), hence the model is highly constrained to data (P12.L19 and P13.L19).

Comment 2: I am confused with "Only the saturated hydraulic conductivity of the top SCL was modified during calibration of the numerical model.? " Why "$\theta r$ $\theta s$ $\alpha$ (m-1) n " were not modified during calibration of the numerical model?

Response 2: We calibrated the model using the whole-pond infiltration rate data as explained above (Response 1) and in the manuscript (P9.L19). During the calibration process we did modify the hydraulic function parameters of the sandy-clay-loam (SCL) layer (4–6 m), nevertheless, changing these parameters did not yield better calibration in terms of infiltration rates. Note that the top SCL layer was practically saturated during the 2015 MAR event (see the water-content profiles at 4 m depth in Fig. 5c, d) and it also has the lowest value of saturated hydraulic conductivity (Ks) along the sediment

profile (Table 1). These combined factors explain why Ks of the top SCL layer was a key parameter for calibration. The above explanation will be added to the revised manuscript (at P9.L22 in the current manuscript).

Comment 3: P11.L25: "In the laboratory, infiltration column experiments with DSW and sand taken from the pond surface (top 0.4 m) showed a reduction by a factor of 1.5 compared to the initial infiltration rate due to compaction-clogging (data not shown)." I think this sentence is not closely related to the above passage.

Response 3: We agree with the reviewer that the context of this sentence as appears in the manuscript is not perfect. The reasoning of this sentence (P11.L25) was to provide lab-scale results that support our field-scale assumption that surface clogging during infiltration with DSW is negligible. We think that reporting these results contributes to the discussion section on clogging, however we will rewrite the paragraph to make sure its context is clear.

Please also note the supplement to this comment:
http://www.hydrol-earth-syst-sci-discuss.net/hess-2016-613/hess-2016-613-AC1-supplement.pdf

---

## Author Response (AR1)

**We would like to thank the Editor for handling the manuscript, and to thank the Referees for their insightful comments, which have helped improving the manuscript. We provide below our**

5 **detailed response to each comment (highlighted). All page and lines numbers refer to the revised marked manuscript.**

**Anonymous Referee #1**

10 This manuscript describes a field experiment conducted and numerical exercises used to evaluate infiltration-recharge dynamics of managed aquifer recharge with desalinated seawater. While their efforts on the field experiments are admirable and the topic is of great interest to the reader of Hydrology and earth System Sciences. I am not good for managed aquifer recharge with desalinated seawater. However, I have a few comments which hopefully might be helpful for further improvement

15 of the paper.

We would like to thank Anonymous Referee #1 for his review and comments. We provide below our detailed response to each comment.

I noticed that the authors' usage of calibration and validation. "The model calibration shows good fit

20 for 90% of the infiltration period (4–31 January 2015) with a relative root mean square error of 4.8% (Fig. 6a)." There are only five points in Fig. 6a, and five points in Fig. 6c?

Response: The numerical model in this study was calibrated to whole-pond data in order to capture the infiltration dynamics of the whole-pond infiltration event (Page 10, Lines 22-23). Most models are calibrated against *head* data, but here we were fortunate to be able to calibrate the model to *flux* data

25 (five points) which usually is unavailable. Other methods for estimating infiltration rates are point-specific (usually indirect) and therefore were not suitable for this purpose. The numerical model was calibrated (and validated) using whole-pond infiltration rates (fluxes) that were calculated by linear-regression of the 5-minutes-resolution ponding-depth data. Two conditions must be met in order to

calculate infiltration rates by this method: (1) ponding depth is declining solely due to infiltration (i.e., no other inlet\outlet source or surface flow) and (2) time span of the descending ponding-depth data is sufficiently long (usually at least few hours) in order to obtain regression with low-error slope (which is a good estimate of the integrated pond-infiltration-rate). Condition (1) was the limiting factor during the operative MAR events and therefore we obtained only five observation points for each calibration and validation. Yet, we emphasize that each observation point was calculated from a large number (tens to hundreds) of ponding-depth data measurements. We revised the manuscript to include the above explanation (Section 2.4.1, Page 4, Lines 20-27). It should also be acknowledged, that unlike most field-scale flow models, here for the top and bottom boundary conditions we used the continuously-measured head values (thousands of measurements), hence the model is highly constrained to data (Page 7, Lines 29-33).

I am confused with "Only the saturated hydraulic conductivity of the top SCL was modified during calibration of the numerical model.? " Why "$\theta r$ $\theta s$ $\alpha$ (m-1) n " were not modified during calibration of the numerical model?

Response: We calibrated the model using the whole-pond infiltration rate data, as explained in our response above and in the revised manuscript. During this calibration process, we tested the effect of the hydraulic function parameters of the sandy-clay-loam (SCL) layer (4–6 m), using the built-in HYDRUS-1D inverse modeling. We found that while changing these parameters improved the fit to the volumetric water content (WC) data, it diminishes it in terms of infiltration rates (underestimating them, hence also the cumulative infiltration). We therefore calibrate the model using the saturated hydraulic conductivity ($K_s$) only, which is sufficient due to the fact that the top SCL layer, which has the lowest value of $K_s$ along the sediment profile (Table 1), was practically saturated during the 2015 MAR event (see the WC profiles at 4 m depth in Fig. 5c, d). We revised the manuscript to include this information, by providing the inverse modeling results altering the hydraulic function parameters, that were eventually not used in the calibrated model (Fig. 6b, also discussed in Page 11, Lines 1-4).

Page 11, Line 25: "In the laboratory, infiltration column experiments with DSW and sand taken from the pond surface (top 0.4 m) showed a reduction by a factor of 1.5 compared to the initial infiltration rate due to compaction-clogging (data not shown)." I think this sentence is not closely related to the above passage.

Response: Following the reviewer's comment we have revised this so that it is better related to that passage (Page 13, Lines 1-6). We note that the information provided in this sentence—lab-scale results that support our field-scale assumption that surface clogging during infiltration with DSW is negligible—contributes to the discussion section on clogging. Therefore, we chose to revise this sentence rather than remove it. In addition, we added the column experiment results to the Supplement (section S2).

**Anonymous Referee #2**

Comments to authors: Based on the solid monitoring and modeling work, the authors are trying to explain the relationship between surface infiltration and groundwater recharge during MAR with desalinated seawater in one pond located in the coastal Israel. Currently, there is few researches done on MAR with DSW in the coastal areas of the world. They obtained a good dataset. However, the authors did not give good illustration on these data and find the main focus of the paper. More consideration as to how this study can inform others elsewhere in the world. I hope that the comments are useful to the authors in revising this study.

Response: We thank the Reviewer for pointing out the uniqueness of our data, and for suggesting improvements to the presentation of our results. We believe our study is relevant to scientists and engineers worldwide as this is the first report (to our best knowledge) on large-scale MAR with desalinated seawater to a Mediterranean coastal aquifer. We hope that the topics discussed in this manuscript will contribute to MAR practitioners and may initiate further research in this direction, as the use of desalinated water as a potable water source is increasing worldwide.

Specific comments:

Page 4 Line 8: "is" should be 'are".

Response: Corrected.

5   2.4 section: Page 4: How to calculate infiltration rates for different conditions should be introduced here. For pond scale, the pond infiltration rates were calculated by linear-regression of ponding-depth declines due to breaks in inlet discharge during MAR event. For single-ring scale, the single-ring infiltration rates were calculated by linear-regression of pressure-head vs. time data. Why did you define that "Infiltration tests with relative errors higher than 15% were omitted from the analysis."? For

10  vadose zone: Infiltration rates were estimated from the velocity of the fronts multiplied by the change in volumetric water content data. For the three aspects mentioned above, could you give the corresponding references or clear conceptual sketch map to illustrate this parameter? This will help readers better understand the infiltration rates under different conditions.

Response: we revised section 2.4 as suggested. Now it includes a detailed explanation on each method

15  for estimating the infiltration-rate, including relevant reference (Section 2.4, Page 4, Line 16 to Page 5, Line 18).

Page 4, Lines 29-31. As the numerical model used both the water table and ponding depth data, what is purpose of the two analytical model under simplified condition?

20  Response: The analytical methods provide a reasonable estimation of the infiltration capacity of the sediment below the pond, which is a valuable approximation for MAR practitioners especially when enough data to calibrate a numerical model is unavailable. It can also serve as a first approximation to verify the results of numerical models. We have revised the manuscript to emphasize this point (Page 5, Lines 22-24 and Page 13, Lines 16-18).

Page 5, Lines 14. The impacts of the assumption of the unit gradient flow for steady state should be evaluated under the transient condition as the water table rise significantly during the recharge period.

Response: Following the Reviewer's suggestion, we revise the manuscript to include an evaluation of the unit gradient assumption using the results of the calibrated numerical model. This discussion appears in section 4.3 (Page 13, Line 30 to Page 14, Line 2), and presented in a new figure (Fig. 8a).

Page 6, Line 32-33. It is not very clear to me how the variable head boundary was applied at the bottom boundary of the Hydrus-1D model and how simulate the transient behavior of the water table. Did you specify variable head at a fixed location and output the flux at this boundary? Please clarify this.

Response: The measured head from the observation well was applied as a boundary condition at a fixed location at the bottom of the model domain. This boundary condition was updated at a one-hour resolution. The recharge flux was the numerical-model output flux at the (fluctuating) groundwater table. Practically, the flux at the groundwater table is the same as the flux at the bottom of the domain because in HYDRUS-1D the variably –saturated flow equation is solved excluding a storage term. We added this clarification to the revised manuscript (Page 7, Line 30 to Page 8, Line 3).

Page 10, Lines 12-14. Another reason why the numerical model seems more reliable than the simplified model may be that the numerical model was calibrated based on observations. I think some testing work of the assumptions (i.e., saturated condition under the pond in the) using the flux distribution from the HYDRUS may be useful for the comparison of the simplified model and numerical model.

Response: The reviewer correctly points out that the numerical model is calibrated, while the simple models are not. For this reason, the simple models are useful for estimating infiltration/recharge rates based on limited data and without the need of calibration procedures. As the reviewer suggested, we used HYDRUS to test the underlying assumptions of the simplified models (Page 13, Line 30 to Page 14, Line 2; Fig. 8a). In addition, Fig. 9 (Fig. 8 in original submission) shows saturated conditions for about 2/3 of the ponding period (i.e., constant water storage in the unsaturated zone), implying that the saturated assumption is valid during most of the ponding period.

Page 11: 4.2 Clogging Line 21-24: Do you have the hydrochemical data to provide evidence for the dissolution of carbonate and gypsiferous matrix?

Response: We did not find evidence for clogging or dissolution as explained in the original manuscript, and therefore revise it to emphasize this point (Page 12, Line 31-33). Nonetheless, we note that pore-clogging with fines originating from the dissolution of carbonate and gypsiferous matrix was suggested by Mukhopadhyay et al. (2004) to explain the observed clogging by desalinated seawater in their column experiments.

Page 12 Line 15-25: this part should be shifted to Methods section.

Response: We have revised the text following this comment, together with the next one; as explained in our next response.

This manuscript estimated groundwater recharge using one dimensional (1D) analytical models and 1D numerical model. The reasons of using 1D models presented in the manuscript is not justified. It is better to present the results using 2D or 3D models, and compare the results 1D models.

Response: We revise the manuscript, altering the discussion section and adding relevant references to justify the use of 1D models (Page 14, Lines 4-6). This assumption is also discussed in the Methods sections. We also added a comparison between the observed groundwater level rise to the results of an analytical model that assumes 1D vertical flow from the pond surface to the groundwater table and radial flow in the aquifer (Page 14, Lines 6-13). The reasonable fit between the observed and modeled groundwater levels supports our 1D flow assumption, as presented in the newly added Fig. 8b.

As was indicated by Zhang et al (2016) that using soil pedotransfer functions directly may lead to biased estimation of dynamic soil moisture content, it is necessary to use site specific pedotransfer functions or inverse models to improve the soil hydraulic parameters used in both analytical models and numerical models. Assouline and Or (2013) also wrote that "The empirical-correlative approach at the basis of the PTF offers reasonable initial estimates for certain large-scale analyses (Romano, 2004). However, the limited physical basis for the estimates of WRC and applicability within the range of values used for the regression analysis, necessitate extra caution for their general application". The

authors may use other pedotransfer functions to compare the results by using Rosetta presented in the manuscript.

Response: During the calibration process, we did modify the hydraulic functions of the sediments by direct and inverse modeling. Nevertheless, changing these parameters did not yield better calibration in terms of infiltration rates. This is because we calibrated the model to the whole pond infiltration rate (Page 10, Lines 22-23, Fig. 6a) while the water content fitting represent point specific data (Fig. 6b). A better fit to the water content was achieved during the calibration process, but the tradeoff was inaccurate whole-pond infiltration rates. We revise the manuscript to include this explanation, including both text (Page 11, Lines 1-4) and graphics (Fig. 6b).

The local sediments of the Israeli Coastal aquifer show reasonable fit between the measured (lab experiments) and estimated saturated hydraulic conductivity as calculated by the pedotransfer functions of the ROSETTA database (Levi, 2015; Shapira, 2012). This contributes to the relatively good performance of all the models used in this study. The revised manuscript now includes this explanation (Page 7, Lines 22-24). Finally, following the reviewer's suggestion, we have added (in Page 7, Lines 24-26) two more tests for our model, using two additional pedotransfer functions (Carsel and Parrish, 1988; Tóth et al., 2015). The revised manuscript includes these test results (Page 11, Lines 8-11, and Fig. S1 in the Supplement).

[revised manuscript text omitted]

---

## Referee Report (RR1)

The revised paper basically well explained the questions from refrees. Another two qeustions in the abstract:

Line 17-20: This sentence is too long to clearly illustrate the reason of clogging. Meanwhile, high-quality is an abscure description of water quality, especially when it is highly related to the important phenomenon-clogging.

Line 21-25: It is stated that numerical simulation can simulate the temporal reduction of infiltration rates. After tracing the detailed information in article, it seems the infiltration is an input for numerical model instead of an output variable. Thus, how the model capture the infiltration rates?

Page9-20, Secondly—Second

---

## Author Response (AR2)

**Letter of response (hess-2016-613), version 2 (minor revision)**

**We would like to thank the Editor for handling the manuscript, and the Referees for their positive and constructive review. We provide below our response to each comment (highlighted). All page and lines numbers refer to the revised marked manuscript.**

**Anonymous Referee #3**

**Submitted on: 26 July 2017**

The revised paper basically well explained the questions from referees. Another two questions in the abstract:

Thanks, detailed response follows each comment.

Comment 1) Line 17-20: This sentence is too long to clearly illustrate the reason of clogging. Meanwhile, high-quality is an abscure description of water quality, especially when it is highly related to the important phenomenon-clogging.

Response 1) The sentence was shortened and revised. Typical turbidity and total dissolved solids of the desalinated seawater that are used for managed aquifer recharge (MAR) were added. These values are small compared to other types of water commonly used for MAR (e.g. treated wastewater, flood water) in which surface clogging is a major concern (Page 2, Lines 18-20; Page 13, Lines 11-12).

Comment 2) Line 21-25: It is stated that numerical simulation can simulate the temporal reduction of infiltration rates. After tracing the detailed information in article, it seems the infiltration is an input for numerical model instead of an output variable. Thus, how the model capture the infiltration rates?

Response 2) The model was calibrated using infiltration-rate data. However the model continuous input at the top boundary was the measured ponding depth (i.e., variable-head BC) during the MAR events and the model outputs is the infiltration and recharge rates (see Figure 6, and explanation on Page 8, Lines 24-28; Page 11, Lines 28-29).

Comment 3) Page9-20, Secondly—Second

Response 3) Secondly was changed to Second as suggested (Page 3, Line 20).

[revised manuscript text omitted]